# Modified Structural and Functional Properties of Fish Gelatin by Glycosylation with Galacto-Oligosaccharides

**DOI:** 10.3390/foods12152828

**Published:** 2023-07-26

**Authors:** Yong Wang, Caiyun Wu, Hui Jia, Jan Mráz, Ran Zhao, Shengjie Li, Xiuping Dong, Jinfeng Pan

**Affiliations:** 1State Key Laboratory of Marine Food Processing and Safety Control, National Engineering Research Center for Seafood, Collaborative Innovation Center of Provincial and Ministerial Co-Construction for Seafood Deep Processing, Liaoning Province Collaborative Innovation Center for Marine Food Deep Processing, College of Food Science and Technology, Dalian Polytechnic University, Dalian 116034, China; wangyong971029@163.com (Y.W.); wucaiyun0722@163.com (C.W.); zr18018984036@163.com (R.Z.); lishj@dlpu.edu.cn (S.L.); dxiuping@163.com (X.D.); 2Institute of Aquaculture and Protection of Waters, South Bohemian Research Center of Aquaculture and Biodiversity of Hydrocenoses, Faculty of Fisheries and Protection of Waters, University of South Bohemia in České Budějovice, 370 05 České Budějovice, Czech Republic; hjia@frov.jcu.cz (H.J.); jmraz@frov.jcu.cz (J.M.)

**Keywords:** glycosylation, fish gelatin, functional properties, structural properties

## Abstract

This study aimed to investigate the effects of galacto-oligogalactose (GOS) glycosylation on the structural and functional properties of fish gelatin (FG). Results showed that with the increase of glycosylation time, grafting degree and browning increased, and new protein bands with increased molecular weight were observed by SDS-PAGE. Structural analysis showed that glycosylation reduced intrinsic fluorescence intensity and increased surface hydrophobicity of FG. FTIR analysis showed α-helix content decreased while random coil content increased in glycosylated FG. Emulsion activity index and emulsion stability index along with foam activity and foam stability were significantly elevated in GOS-4 and GOS-8, but FG glycosylated longer than 12 h exhibited less pronounced improvement. Glycosylated FG showed lower gel strength than control. The results indicate that moderate glycosylation could be applied to improve interfacial properties of FG.

## 1. Introduction

Gelatin is widely used in the food industry as a versatile polymeric biopolymer with good gelling, emulsifying, and film-forming properties [1,2]. Currently, commercial gelatin is mainly produced from mammals, such as the bones and skins of cattle and pigs. However, mammalian diseases and religious and cultural beliefs have limited the use of mammalian gelatin in food products [3]. Compared to mammalian gelatin, fish gelatin (FG) is becoming an excellent alternative to mammalian gelatin due to its high safety, unrestricted application, and high exploitation value. However, the low content of amino acid (proline and hydroxyproline) in FG leads to poor gelling and emulsifying properties, hindering its application in foods [4,5]. There is an urgent need to modify FG to improve its functional properties to meet the needs of the food industry.

FG can be modified by enzymes [6,7], chemicals (phosphorylation, glycosylation, phenolics, and aldehydes, etc.) [8,9,10,11], electrolytes or non-electrolytes [12,13], and physical treatments (ultrasound, irradiation, high pressure) [14,15]. Usually, chemical modification of protein is more stable and effective compared with other strategies. Glycosylation is an effective chemical method for protein modification, which mainly involves the covalent binding of ε-amino groups in protein molecules with carbonyl groups in reducing sugars to generate protein–polysaccharide adducts, thus changing the functional properties (emulsifying, foaming) of proteins [16]. Studies have shown that glycosylation can alter the functional properties of FG, especially for improving interfacial properties [9,17,18]. Zhao et al. [9] and Guo et al. [18] used gum Arabic and glucose to glycosylate FG, respectively, which both improved the emulsification of the gelatin solution but reduced the gelling properties. Currently, most studies used reducing sugars with simple structures and limited nutritional and functional properties, such as glucose, mannose, fructose, and gum Arabic for protein glycosylation while polysaccharides with functional properties are rarely selected. Galacto-oligogalactose (GOS), a reducing polysaccharide, is a novel functional prebiotic with a series of physiological functions such as promoting the proliferation of intestinal probiotics, improving mineral absorption, lowering cholesterol, and promoting the synthesis of nutrients in the body [19,20]. The glycosylation of FG with GOS might not only lead to the elevation of gelling or emulsifying properties but could also bring nutritional value improvement to FG, which could increase the commercial value of FG. Even though, there are many previous studies which revelated that glycosylation can improve FG properties. However, studies using galacto-oligosaccharides glycosylated FG have not been found, and there is still lacking information related to in-depth structural changes of them.

This study aimed to evaluate the effects of glycosylation of FG with GOS on the structural and functional properties of FG. The changes in functional properties such as emulsifying, foaming, and gelling were investigated. Changes in protein pattern were analyzed by sodium dodecyl sulfate polyacrylamide gel electrophoresis (SDS-PAGE), and the structural properties were studied by fluorescence spectroscopy, Fourier transform infrared (FTIR), and circular dichroism (CD) spectrum. The results might provide beneficial knowledge for applying glycosylation to improve FG properties.

## 2. Materials and Methods

### 2.1. Materials

FG was purchased from Shanghai Yuanye Bio-Technology Co., Ltd. (Shanghai, China). Galacto-oligosaccharides (GOS) were purchased from Solarbio Science & Technology Co., Ltd. (Beijing, China). All other chemicals were of analytical grade and purchased from Sangon Biotech Co., Ltd. (Shanghai, China).

### 2.2. Preparation of Glycosylated FG

Glycosylated FG was performed according to the method of Wang et al. [21] with modifications. FG powder was dissolved in 40 °C deionized water with sufficient stirring to form a 4% FG solution. GOS powder was added into the FG solution to be fully dissolved. pH was adjusted to 8 with 1 M NaOH to obtain FG–GOS solution where the mass ratio of FG to GOS was 1:1. The FG–GOS solution was freeze-dried after pre-freezing at −20 °C to obtain dried FG–GOS samples. FG–GOS sample was ground into powder and placed in an oven with temperature of 50 °C and humidity of 75% (provided by saturated NaCl solution). Glycosylation reactions were carried out for 0 h, 4 h, 8 h, 12 h, 16 h, and 20 h. When the reactions were completed, samples were transferred into a −20 °C refrigerator to terminate the reactions. The glycosylated FG were named GOS-0, GOS-4, GOS-8, GOS-12, GOS-16, and GOS-20, respectively. The unglycosylated FG was control (CO).

### 2.3. Analysis of Grafting Degree

The grafting degree (GD) of glycosylated FG was determined using the O-phthalaldehyde (OPA) method according to Bostar et al. [22] with modifications. OPA solution was freshly prepared. OPA (40 mg) was dissolved in 1 mL of ethanol, 2.5 mL of 20% (*w*/*w*) SDS, and 100 μL of β-mercaptoethanol was added and diluted to 50 mL with 0.1 M Borax buffer (pH 8.5). FG (*w*/*v* = 2%, 200 μL) was added into 4 mL of OPA solution and the absorbance at 340 nm was measured using an UV 5200B spectrophotometer (Shanghai Metash Instrument Co., Ltd., Shanghai, China). GD was calculated as follows:(1)$$\mathrm{GD}\;\left(\%\right)=\frac{A_{0}-A_{t}}{A_{0}}\times 100\%$$
where A_0_ and A_t_ in the formula were the absorbance of sample before and after glycosylation.

### 2.4. Browning Degree Analysis

The browning degree of glycosylated FG was determined according to the method of Ai et al. [23]. FG was dissolved and the concentration was adjusted to 2 mg/mL, and the absorbance at 420 nm was measured using a spectrophotometer. The dried and gel samples of glycosylated FG were also photographed and observed using a digital camera.

### 2.5. FG Structure Analysis

#### 2.5.1. Protein Patterns

Protein patterns of glycosylated FG were analyzed using SDS-PAGE according to Pan et al. [24]. FG was diluted and protein concentration was adjusted to 5 mg/mL. The diluted sample was mixed with 5× loading buffer containing 5% β-mercaptoethanol, 8 mol/L Urea, 5% SDS (*m*/*v*), 0.25 mol/L pH 7.5 Tris-HCl, and bromophenol blue at a ratio of 1:1 (*v*/*v*). Samples were boiled for 5 min and centrifuged. The supernatant was collected as a sample for electrophoresis. The conditions for electrophoresis were as follows: 5% concentrated gel, 12% separation gel, 15 μL sample, 30 mA constant current. The gels were stained with 0.25% Coomassie blue and decolorized with a mixture of methanol and acetic acid.

#### 2.5.2. Surface Hydrophobicity

Surface hydrophobicity of glycosylated FG was determined according to Pan et al. [25] with modifications. FG solution (1 mL, 2.5 mg/mL) was shaken with 200 μL of 1 mg/mL bromophenol blue for 10 min at room temperature with aluminum foil to avoid light. After centrifugation (4 °C, 4000 r/min, 15 min), 1 mL of supernatant was aspirated and diluted 10-fold. The absorbance of sample was measured at 595 nm. Hydrophobicity was calculated as the following equation:(2)$$\mathrm{BPB}\;\mathrm{bound}/\mu g=\frac{200\;\mu g\;\times \;\left(A_{\mathrm{Control}}-A_{\mathrm{Sample}}\right)}{A_{\mathrm{Control}}}\;\;$$
where A_Control_ = absorbance of supernatant without sample but using deionized water as control; A_sample_ = absorbance of supernatant containing FG sample.

#### 2.5.3. Intrinsic Fluorescence

The intrinsic fluorescence of glycosylated FG was determined by fluorescence spectrometry (F2700, Hitachi Co., Tokyo, Japan) according to Huang et al. [26] with modifications. The analysis conditions were as follows: sample concentration 1% (*w*/*v*), excitation wavelength 280 nm, emission wavelength 290–480 nm, scan speed 1500 nm/min, slit width 5 nm.

#### 2.5.4. Fourier Transform Infrared

The structural properties of glycosylated FG were analyzed by FTIR spectroscopy (Perkin Elmer, Inc., Madison, WI, USA) according to the method of Wang et al. [27]. Glycosylated FG was ground and mixed with KBr at a ratio of 1:100 (*w*/*w*) and pressed into thin slices for measurement. The spectra were collected in the range of 400–4000 cm^−1^ and averaged over 32 scans at a resolution of 4 cm^−1^.

#### 2.5.5. Circular Dichroism (CD) Spectrum

The secondary structure of glycosylated FG was determined by circular dichroism spectrometry (J1500, JASCO, Tokyo, Japan) using the method of Zhao et al. [9] with modifications. Sample concentration was diluted to 1 mg/mL and the spectra at a wavelength range of 250–200 nm was analyzed. The obtained data were used to calculate the ratio of the secondary structure (α-helix, β-sheet, β-turn, and random coil) of FG using the attached software with the instrument.

### 2.6. Emulsifying Properties Analysis

#### 2.6.1. Emulsion Activity Index (EAI) and Emulsion Stability Index (ESI)

EAI and ESI of gelatin were determined according to Liu et al. [28] with modifications. Briefly, soybean oil (2 mL) and gelatin solutions (1 g/L, 6 mL) were homogenized at 13,000 rpm for 1 min. The emulsion (100 μL) was taken at 0 and 10 min, and diluted with 0.1% SDS solution (×400). The absorbance of emulsion at 500 nm was measured using a spectrophotometer. EAI and ESI were expressed as:(3)$$\mathrm{EAI}\left(m^{2}/g\right)=2\times 2.303\times \frac{A_{0}\times \mathrm{DF}}{\mathrm{IC}\Phi }$$
(4)$$\mathrm{ESI}\left(\min\right)=\frac{A_{0}\times t}{\Delta A}$$
where A_0_ and A_10_ = absorbance at 500 nm measured at 0 and 10 min after emulsification, DF = dilution factor (100), I = path length of cuvette (0.01 m), Φ = oil volume fraction (0.25), C = protein concentration in the aqueous phase (g/m^3^), t = 10 min, and ΔA = A_0_ − A_10_.

#### 2.6.2. Microscopic Observation of Emulsion Droplet

The droplet morphology of glycosylated FG emulsions was observed by optical microscopy following the method of Niknam et al. [29] with modifications. A mixture of 10 mL of soybean oil and 40 mL of gelatin solution (1%, *w*/*v*) was homogenized at room temperature for 1 min. After continuous stirring for 30 min, the pH of the emulsion was adjusted to 7. One drop of the emulsion was added to a slide, covered with a coverslip, and observed using an OLYMPUS-DP72 optical microscope (Olympus Co., Tokyo, Japan).

### 2.7. Foaming Properties Analysis

The foaming properties of glycosylated FG were determined according to Wang et al. [30]. FG solution (50 mL) with a concentration of 1% (*w*/*v*) was homogenized at 12,000 rpm for 2 min at room temperature, quickly transferred to a 250 mL measuring cylinder, and the foam heights were recorded at 0 min and 30 min. The measurement was triplicated. Foam activity (FA) and foam stability (FS) were calculated as the following equations:(5)$$\mathrm{FC}\;\left(\%\right)=\frac{V_{1}}{V_{0}}\times 100\%$$
(6)$$\mathrm{FS}\;\left(\%\right)=\frac{V_{2}}{V_{1}}\times 100\%$$
where V_0_ = volume of the solution before whipping. V_1_ = foam volume after whipping. V_2_ = foam volume after 30 min.

### 2.8. Gel Strength Analysis

FG solution of 4% was prepared. The solution was dispensed into 5 mL beakers and cooled at 4 °C for 16 h to mature into a gel of 2 cm in diameter and 2.5 cm in height. Gel strength was determined using a TA.XT. Plus texture analyzer (SMS, Surrey, UK) according to Pan et al. [31]. The measuring parameters were as follows: probe selection: P/5, pre-test speed: 1.0 mm/sec, test speed: 0.5 mm/sec, post-test speed: 10 mm/sec, and penetration distance: 4 mm.

### 2.9. Statistical Analysis

Data were reported as mean ± SD. Comparison of means was performed using Duncan’s multiple range tests in the Statistic Package for Social Science (SPSS 16.0, SPSS Inc., Chicago, IL, USA). The significant level was set at *p* < 0.05.

## 3. Results and Discussions

### 3.1. GD and Browning

Glycosylation of proteins with reducing sugars occurs mainly between the free amino groups on the amino acid side chain of the protein and the carbonyl group at the reducing end of the sugar molecule [30]. As shown in Figure 1A, it is found that the GD of FG increased significantly as glycosylation time prolonged, especially before 8 h, and it slowly increased until 20 h. The result indicates that GOS successfully bound to FG molecules, and the rate of glycosylation reaction depended much on time. At the beginning, the active amino groups on the protein surface could react easily with the active carbonyl groups of polysaccharide, and their binding speed might be fast [32]. However, as reaction time extended, the binding ability of FG to GOS decreased due to the structural changes and degradation of the conjugate as well as the saturation of the glycosylation site [33], leading to a slow increase of GD.

The process of glycosylation was accompanied with the formation of brownish materials [23]. As shown in Figure 1B, no obvious browning was observed until 4 h glycosylation. Nevertheless, brownness of the sample increased dramatically after 8 h and it kept fast increasing until 20 h when the most obvious browning was observed. Figure 1C exhibits the color change of FG glycosylated at different times, which was consistent with the changes of browning. The color gradually changed from white to yellow-brown after 8 h glycosylation, and GOS-16 and GOS-20 showed strong yellowness. Generally, browning degree is highly correlated with GD. It is considered that a high level of grafting suggests the sample is at the late stage of Maillard reaction which would produce more pigment substances [34]. However, heavy darkness would lower the sensory quality of the final product, and it might also reduce the nutritional value of protein [30]. Thus, a suitable reaction time should be selected for glycosylation to obtain FG with both good functional properties and nutritional properties.

### 3.2. SDS-PAGE

The results of SDS-PAGE analysis of glycosylated FG is shown in Figure 2. It is found that the protein pattern of GOS-0 and GOS-4 showed no change compared with that of CO, indicating the glycosylation of FG did not occur deeply with heating less than 4 h. However, the band with molecular weight close to 135 KDa was greatly attenuated along with enhanced intensity of bands at the top of the separation-gel in GOS-8, GOS-12, GOS-16, and GOS-20. Protein side chains of FG might generate disulfide bonds or isopeptide-bonded glycans with polysaccharide to form glycoproteins during glycosylation [35], leading to increased FG molecular weight. Results indicate that FG could be successfully glycosylated since 8 h, confirming the conclusion supported by results of GD and browning. Noticeably, GOS-12, GOS-16, and GOS-20 also exhibited some “ghost bands” in the concentration-gel area, which could be due to the formation of insoluble high-molecular- weight FG–GOS adducts when extremely heavy grafting occurred. Gelatin are combination hydrolysate polymer chains with different molecular weights including α-chains (80~125 KDa), β-chains (160~250 KDa), γ-chains (240~375 KDa), and other low-molecular substances [4]. The distribution of these chains affects the functional properties of gelatin, with higher α-chain content resulting in higher gelling properties [36], whereas the generation of insoluble high-molecular-weight proteins may negatively affect gelatin-related functional properties. Therefore, we hypothesized that heavy grafting would lead to the deterioration of gelling properties of FG.

### 3.3. Surface Hydrophobicity and Intrinsic Fluorescence

The surface hydrophobicity of glycosylated FG is shown in Figure 3A. Compared with CO, surface hydrophobicity of GOS-4 and GOS-8 increased significantly, which could be explained by the exposure of hydrophobic groups by heating in the early stage of glycosylation. The grafting of GOS onto FG might also stabilize the hydrophobic groups of gelatin. The slight increase of hydrophobicity in GOS-0 could be induced by the occurring of GOS. Surface hydrophobicity decreased as glycosylation time extended to longer than 8 h. It is possible that heavy glycosylation generate high-molecular-weight polysaccharide-gelatin adducts that might have shielding effect to bury hydrophobic groups, reducing surface hydrophobicity [37]. In addition, heavy glycosylation might bring more hydrophilic hydroxyl groups into FG, which would lead to the decrease of hydrophobicity [23]. Surface hydrophobicity is closely related to emulsification properties and foaming properties. Increased surface hydrophobicity implies the exposure of hydrophobic groups, which allows gelatin molecules to adsorb more rapidly to the oil–water interface to form emulsions, thus facilitating the improvement of emulsification properties and foaming properties of FG.

Intrinsic fluorescence is often measured to indicate the internal structure of protein. As shown in Figure 3B, the fluorescence intensity of FG decreased as glycosylation time extended, suggesting the tertiary conformation of FG changed. The declined intrinsic fluorescence could be attributed to the shielding effect of the polysaccharide molecular chain in the region around the fluorescent moiety, and the spatial site blocking effect by including polysaccharides, likewise, would enhance the shielding effect [38]. Additionally, studies have shown that the fluorescent chromophores of proteins are easily exposed to the solvent environment after glycosylation, which causes fluorescence bursts and leads to a decreased fluorescence intensity [30,39]. The changes in the tertiary structure of FG induced by glycosylation might greatly influence the functional properties of FG.

### 3.4. FTIR and Secondary Structure

FTIR spectra of glycosylated FG were shown in Figure 4A. Generally, an increased hydroxyl content is typical changed when protein is glycosylated, which shows an absorption peak at 3700–3200 cm^−1^ [39]. Compared with CO, glycosylated FG had wider peak at 3700–3200 cm^−1^ with increased intensity, implying the generation of FG–GOS adducts by covalent binding, which would increase the wave number of hydroxyl groups and enhance the C-OH stretching vibration [40]. The amide-I band (1600–1700 cm^−1^) and amide-III band (1200–1300 cm^−1^) were shifted toward higher wavelengths. The amide-I band is the characteristic band of gelatin helical structure [41]. The results indicate that glycosylation was not favorable for the formation of the FG helical structure, which might affect the gelling properties of FG. The amide-II band (1500–1550 cm^−1^) showed no significant change. All glycosylated FG showed the characteristic peaks of GOS whose absorption intensity was the highest in GOS-0 but gradually decreased as glycosylation time prolonged. This should be due to the fact that a high grafting degree is along with the generation of more FG–GOS adducts, which changed the chemical structure of GOS and weakened the signal generated by functional groups of GOS. The result evidences the incorporation of GOS in glycosylated FG.

As shown in Figure 4B, the secondary structure content of glycosylated FG was featured with a high proportion of β-sheet, β-turn, and random coil while having a low level of α-helix. Compared with CO, most glycosylated FG showed increased α-helix and β-turn but decreased random coil. However, α-helix and β-turn proportion of FG decreased along with increased β-sheet and random coil as glycosylation time prolonged from 4 h to 20 h. This suggests that glycosylation causes the conformational transition of FG from ordered to disordered [42]. The change in secondary structure of FG could also be due to thermal treatment during glycosylation [43]. The introduction of GOS reduced the stability of gelatin conformation, especially for the α-helix structure that is crucial for gelling properties, but the flexibility of its conformation may be improved, which might be beneficial for interfacial properties.

### 3.5. Functional Properties of Glycosylated FG

#### 3.5.1. Emulsification Properties

As shown in Figure 5A, glycosylation significantly improved the emulsifying activity of FG. GOS-4 and GOS-8 showed the highest EAI, 11.02 m^2^/g and 11.42 m^2^/g, while the others showed EAI < 10 m^2^/g. It is considered that covalent cross-linking of gelatin molecules with polysaccharides changed the tertiary conformation of gelatin molecules, which improved the surface hydrophobicity and enhanced the adsorption capacity of FG on the oil–water interface [42]. However, when glycosylation was performed for >12 h, some insoluble high-molecular-weight FG–GOS adducts could be formed. Meanwhile, prolonged glycosylation time might cause increased protein denaturation and aggregation to reduce solubility, consequently resulting in deteriorated emulsifying activity [30,44]. Changes of ESI exhibited a similar pattern with that of EAI, whose highest value was observed in GOS-4 and GOS-8, and it declined when glycosylation time was >12 h. It is hypothesized that FG–GOS adducts generated by glycosylation delayed the flocculation of the emulsion droplets by inducing spatial repulsion [9], improving their emulsion stability. Zhang et al. [17] also found that the glycosylation of gelatin with monosaccharides improved the emulsion stability.

The droplet morphology of the glycosylated FG emulsions is shown in Figure 5B. The number of emulsion droplets formed by CO was the lowest while the diameter was the largest. GOS-4 and GOS-8 formed many small-size emulsion droplets, which corresponded to their high ESI. When glycosylation time was further prolonged, the number of emulsion droplets decreased, though the size of emulsion did not change significantly. These suggest that glycosylation could reduce the diameter of the emulsion, increasing the total boundary area of the droplet, which would be beneficial for maintaining emulsion stability. Ai et al. [23] reported that the flexibility of protein increased after glycosylation, and they considered lateral electrostatic and spatial repulsion effects were beneficial to the coverage and stabilization of protein at the oil–water interface when the protein–polysaccharide affixation was adsorbed at the oil–water interface. On the other hand, the inclusion of polysaccharides might change its own spatial site resistance, which was favorable for stabilizing emulsions. The above results indicate that mild glycosylation could improve FG emulsifying properties.

To better understand the effect of glycosylation on emulsifying properties, Figure 6 depicts a proposed mechanism. It is thought that GOS and FG react to generate FG–GOS adducts. In moderate glycosylation, internal hydrophobic groups of FG are exposed, and appropriate amounts of FG–GOS adducts could stabilize the exposed hydrophobic groups. The exposure of hydrophobic groups enabled the FG molecules to absorb more rapidly to the oil–water interface, while the spatial repulsion generated by the FG–GOS adducts retarded the flocculation and aggregation of the emulsion droplets, this “cooperation” improved emulsifying properties of FG. However, heavy glycosylation leads to excessively poor soluble FG–GOS adducts, which buries hydrophobic groups and introduces a large number of hydrophilic hydroxyl groups. Meanwhile, insoluble FG–GOS adducts are not involved in emulsification and cannot be adsorbed to the surface of emulsion droplets, which is not able to stable emulsions.

#### 3.5.2. Foaming Properties

In Figure 7, glycosylated FG for 4–12 h showed higher FC and FS than CO and GOS-0. And GOS-12 and GOS-16 showed lower FC and FS than GOS-4 and GOS-8. Surface hydrophobicity is crucial to the foaming performance of protein. Usually, high surface hydrophobicity represents a high affinity for the air/water interface, which is necessary for maintaining good foaming capability [45]. FG changed its tertiary conformation after short-time glycosylation. It exposed more internal hydrophobic groups (see Figure 3A), resulting in increased surface hydrophobicity. In contrast, overmuch glycosylation might lead to excessive grafting which generates some high-molecular-weight FG–GOS adducts that could compact the conformation of FG [46], resulting in reduced hydrophobicity to provide declined FC and FS.

#### 3.5.3. Gel Properties

Gel strength of glycosylated FG is shown in Figure 8. It is found that only GOS-4 exhibited similar GS with CO, but all other glycosylated FG showed lower gel strength than CO. The results indicate that glycosylation decreased gelling properties of FG. During glycosylation, high-molecular-weight FG–GOS adducts with low solubility would be formed, and this is not conducive to the formation of hydrogen bonds during FG gelation and the formation of dense gel networks [9]. Further, Figure 4B shows the declined amount of α-helix in FG after glycosylation. α-helix is an important secondary structure for gel-forming capability of gelatin. Usually, a high proportion of α-helix could provide stable triple-helix structure to gelatin. The latter could help form a strong gel network by forming more hydrogen bonds. Similar results were reported by Zhao et al. [9] and Huang et al. [47]. Noticeably, GOS-0 had a lower gel strength than CO. It could be that GOS is not bound to FG but would be hydrated during the gelation and competes with the gelatin molecules for water, therefore hindering the formation of a gel network.

## 4. Conclusions

GOS was successfully grafted onto FG by a dry-heating method. The grafting degree and browning degree increased as the glycosylation time was prolonged. Good emulsifying and foaming properties can be obtained after mild glycosylation (4 or 8 h) due to the elevated surface hydrophobicity and enhanced adsorption capacity of FG at the oil–water interface. However, heavy glycosylation (>12 h) might lead to the formation of excessive insoluble FG–GOS adducts or protein denaturation, resulting in weak emulsifying and foaming activity. α-helix content of glycosylated FG decreased while random coil content increased, which lowered the stability of the gelatin molecular conformation and was not conducive to the formation of the gel network, resulting in deteriorated gel properties. Thus, mild glycosylation could be applied for improving FG interfacial properties.

## Figures and Tables

**Figure 1 foods-12-02828-f001:**
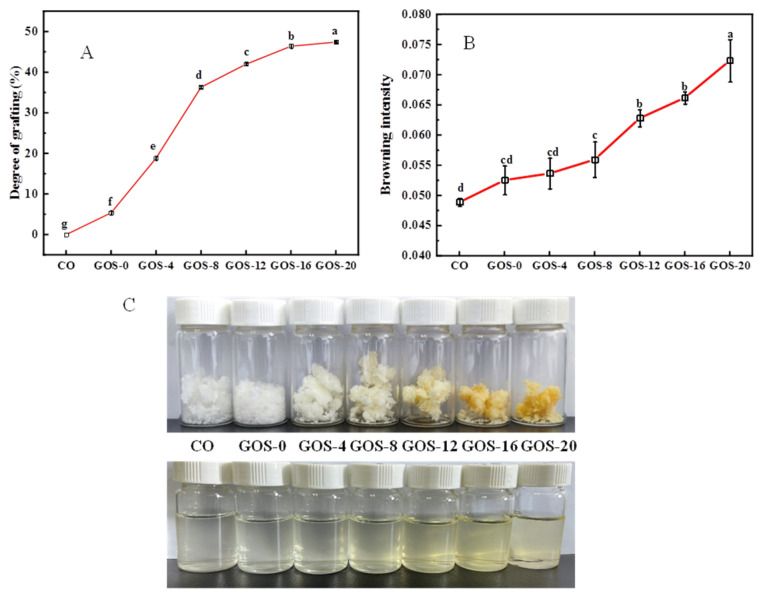
The grafting degree (**A**), browning intensity (**B**), and appearance (**C**) of FG glycosylated for different times. GOS-numbers = glycosylation time; CO = no glycosylation. Lowercase letters indicate significantly different levels between groups (*p* < 0.05).

**Figure 2 foods-12-02828-f002:**
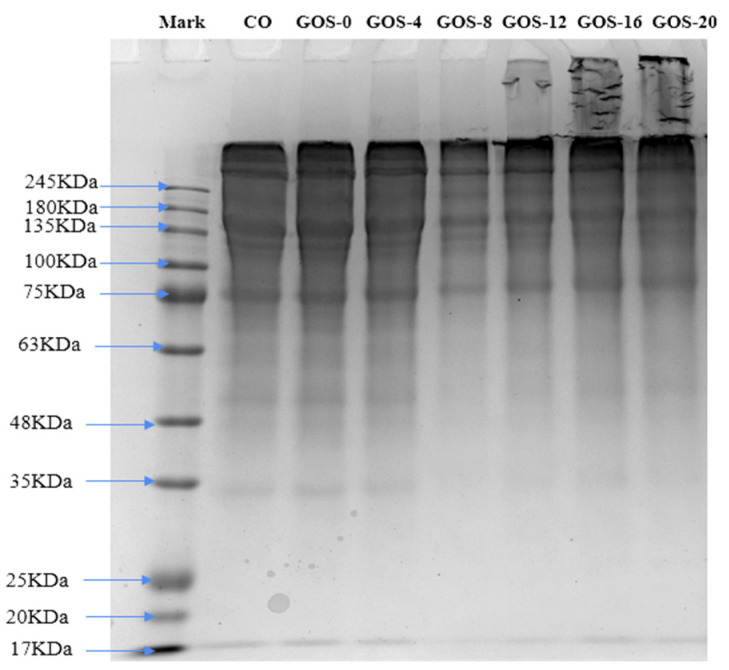
The SDS-PAGE electrophoresis analysis of glycosylated FG.

**Figure 3 foods-12-02828-f003:**
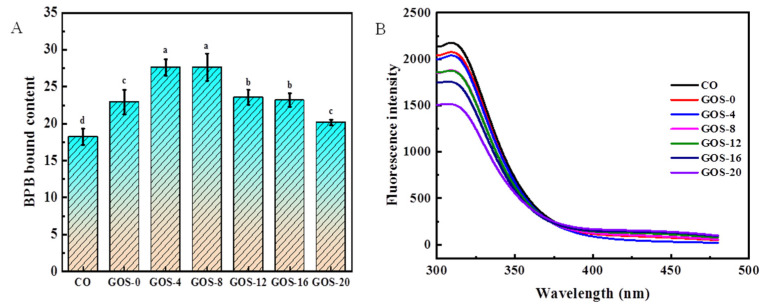
The surface hydrophobicity (**A**) and intrinsic fluorescence (**B**) of glycosylated fish gelatin. Lowercase letters indicate significantly different levels between groups (*p* < 0.05).

**Figure 4 foods-12-02828-f004:**
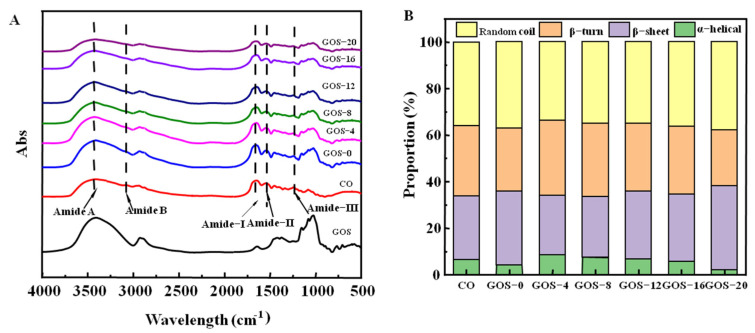
The Fourier transform infrared spectra (**A**) and secondary structure (**B**) of glycosylated fish gelatin.

**Figure 5 foods-12-02828-f005:**
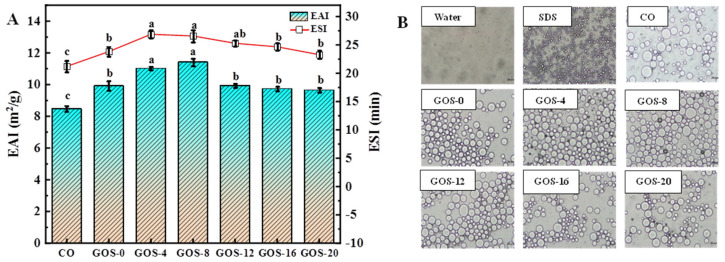
The emulsifying properties (**A**) and emulsion droplet morphology (**B**) of glycosylated fish gelatin. Lowercase letters indicate significantly different levels between groups (*p* < 0.05).

**Figure 6 foods-12-02828-f006:**
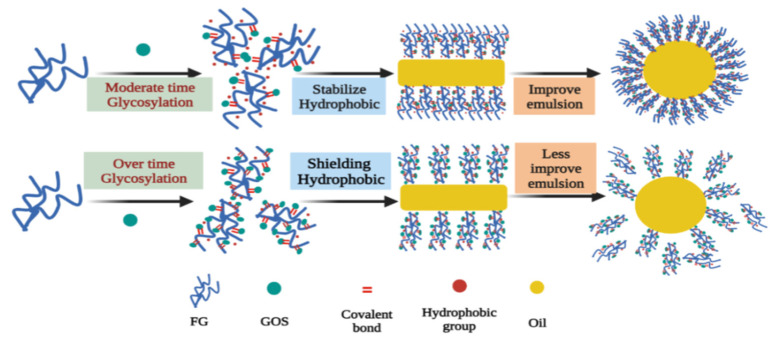
The schematic diagram illustrates the mechanism by which glycosylation affects emulsifying properties of FG.

**Figure 7 foods-12-02828-f007:**
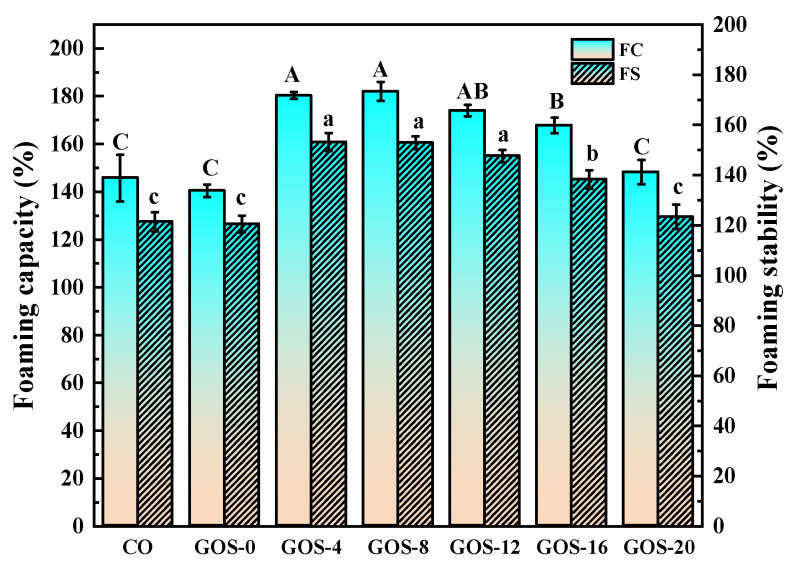
The foaming properties of glycosylated fish gelatin. Lowercase and capital letters indicate significantly different levels between groups (*p* < 0.05).

**Figure 8 foods-12-02828-f008:**
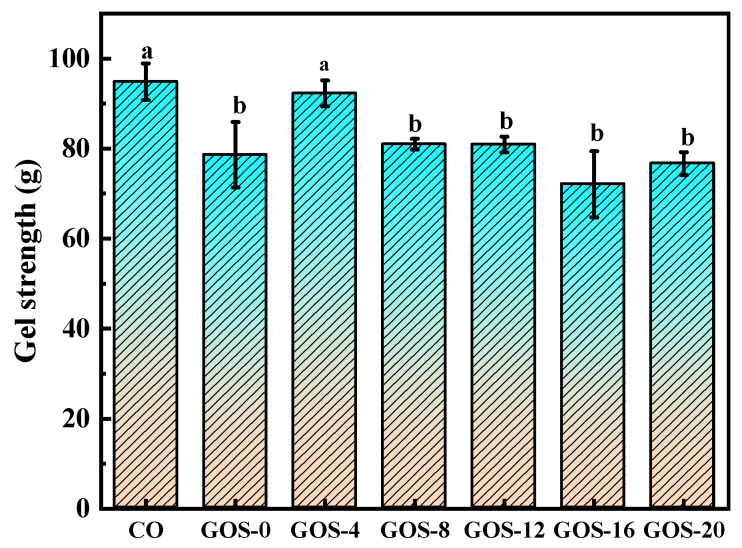
The gel strength of glycosylated fish gelatin. Lowercase and capital letters indicate significantly different levels between groups (*p* < 0.05).

## Data Availability

The data presented in this study are available on request from the corresponding author. The data are not publicly available due to the application for patent.

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
