# Peer review of "Modified Structural and Functional Properties of Fish Gelatin by Glycosylation with Galacto-Oligosaccharides"

_foods, 2023, doi:10.3390/foods12152828_

Round 1
Reviewer 1 Report
This study determines the structural and functional properties of fish gelatin by glycosylation with galacto-oligosaccharides. Overall, the introduction, experimental design, or even English is well-written and acceptable. There are minor concerns asking author to revise as follows:
ABSTRACT
- Line14: Please add the space between “and” and “functional”
INTRODUCTION
- Overall, introduction is well-written, however, it can be improved to strengthen this MS. At line 52-54, author mentioned that glycosylation with GOS might not only improve gelling and emulsifying properties, but also improve nutritional value. Then, the following objectives of this study were addressed. In this issue, the context/fluency is queried. Now, the research gaps and objectives are not comprehensive. I suggest author to add some sentences related to objectives such as “Even there are many previous studies revealed that GOS can improve FG properties, but there is still lacking information related to in-depth structural changes of them,…………” etc.
MATERIALS AND METHODS
-Line68: Please check???
RESULTS AND DISCUSSION
- Line 207: For protein pattern, is there any previous studies also revealed the appearance of protein bands with MW higher than 245 kDa? They should be added. Moreover, how’s these high MW protein affected to gelatin properties should be addressed or discussed.
- Line222: The effect of hydrophobicity on gelatin properties should be added. The high or low hydrophobicity is good? Why?
- Overall, the findings of the study are fairly discussed, yet more references about glycosylation to improve FG or others are needed to enrich the MS.
CONCLUSION
- Line357: GOS was successfully grafted onto FG by dry-heat method??? How’s come with this conclusion? Dry-heat?? Please clarify this point.
- The conclusion seem to be optimize the glycosylation time to improve FG properties? Therefore, the topic is maybe changed.

Reviewer 2 Report
This manuscript is modified structural and functional properties of fish gelatin by glycosylation with galacto-oligosaccharides. It is interesting and I think, after minor revision of manuscript, it can be considered. You can find my comments in below:
1. The manuscript must be revised grammatically and the English level of it must be improved by a native editor.
2. The authors must re-write the abstract and conclusion sections. I think some sentences are not needed to be in these sections.
3. In introduction section, please add more references about the same researches which worked on current research title. Authors must write them in the last paragraph of introduction.
4. Please add appropriate references for sections 2.2 and 2.5.2.
5. Please add appropriate references for section 2.6 (Emulsifying properties) like https://doi.org/10.1016/j.foodhyd.2022.107758.
6. In section results and discussion, please add more details about the obtained data. Also, authors must more compare the obtained results with the results of previous researches.
7. Please increase the DPI values of figures. The quality of them is poor.
The manuscript must be revised grammatically and the English level of it must be improved by a native editor.
